# Green Human Resource Management/Supply Chain Management/Regulation and Legislation and Their Effects on Sustainable Development Goals in Jordan

Lana Freihat [1], Mousa Al-Qaaida [2], Zayed Huneiti [3] and Maysam Abbod [1,*]

1   Department of Electronic and Electrical Engineering, College of Engineering, Design and Physical Sciences, Brunel University London, Uxbridge UB8 3PH, UK; lanajamilsh.freihat@brunel.ac.uk
2   Department of Law, Faculty of Law, Amman Arab University, Jordan Street–Mubis, Amman 11953, Jordan; m.alqaaida@aau.edu.jo
3   Electrical Engineering Department, Faculty of Engineering Technology, Al-Balqa Applied University, Amman 11134, Jordan; zayedhuneiti@bau.edu.jo
*   Correspondence: maysam.abbod@brunel.ac.uk

**Abstract:** In recent decades, sustainability and environmental concerns have become increasingly significant topics of discussion. This article aims to propose a conceptual framework of a research model including the correlations between government regulations and legislations, Green Human Resource Management (GHRM), Green Supply Chain Management (GSCM), and Sustainable Development Goals (SDGs). The methodological approach adopted in this study included conducting a review of the relevant literature and accessing databases and search engines to gather information. The current article presents a novel approach to understanding how organizations and regulators can collaborate to drive sustainable development in this domain. This study also adds significant value due to its unique contribution in connecting GHRM, GSCM, and government regulation and legislation, particularly in the context of sustainable development and its link to promoting decent work and economic growth (SDG8), responsible consumption and production (SDG12), and addressing climate action (SDG13). The rarity of articles addressing these interrelated topics, especially within the specific context of Jordan, where such research has been largely absent, underscores the distinctive nature of this study. Furthermore, this article stands out for its comprehensive incorporation of legal and regulatory aspects into the discourse on organizational GHRM and GSCM practices and their alignment with the pursuit of SDGs. By providing valuable insights for decision makers and organizations, including a thorough examination of the barriers involved, this article serves as an essential resource for understanding and navigating the complex interplay between environmental sustainability, GHRM, GSCM, and governmental regulations. Based on the analysis of the findings, a conceptual framework is proposed based on three environmental dimensions and six green practices that have discernible effects. Finally, it is envisaged that this study will offer directions for future research work to use another approach and another environment.

**Keywords:** green human resource management; green supply chain management; regulation and legislation; sustainable development goals

## 1. Introduction

Sustainability has become increasingly important to societies on a global scale, with growing concern about the viability and effectiveness of resources, infrastructure, policies, and economies amidst challenges such as climate change, over-consumption, and global population growth. Since the term "sustainability" was popularized from its policy-based origins through the Brundtland report, the term "sustainability" has been cast to have a wide range of meanings, evolving from its initial emphasis on the eco-environmental

aspects of development concerning the environment to a broader context encompassing three pillars: social, economic, and environmental sustainability [1–3].

Jordan, aligned with this global trend, has rooted its national development strategies, plans, and policies within the framework of the 2030 Agenda, encompassing its goals and targets. Notably, Jordan's current development plan, the Government's Indicative Executive Program (GIEP), stands out as the most ambitious to date, as it emphasizes the integration of the Sustainable Developments Goals (SDGs). Jordan has also taken a step to mainstream the SDGs into various sectoral strategies and action plans, thereby complementing the efforts outlined in the GIEP. One example is the integration of the SDGs into Jordan's Green Growth National Action Plan 2021–2025. [2,4–6]. Four principles driving green growth have been identified and mainstreamed throughout the actions in the Green Growth National Action Plan 2021–2025; these principles include transparent government processes, regulations, and legislation, fostering a behavioral shift and capacity building (creating new green jobs), implementing mechanisms for stimulating green growth, and adopting integrated planning processes that value societal impacts [6].

GHRM, GSCM, and regulation and legislation play a crucial role in achieving SDGs in Jordan. Jordan's Second Voluntary National Review (VNR) in 2022 highlighted the country's commitment to all 17 SDGs and the progress made across different goals, reflecting the interlinkages and synergies between them [4]. Stephen Early IV, Lawrence (2020) emphasizes the importance of regulation in promoting sustainability by addressing the linkages between social, environmental, and economic factors [1]. This is particularly relevant in the context of GHRM and GSCM, where regulatory frameworks play a significant role in shaping organizational practices and their impact on sustainability.

Several studies have emphasized the interrelation between GHRM, GSCM, and sustainable development. For instance, a study by Chiappetta Jabbour and Sousa Jabbour (2016) [7] linked GHRM and GSCM, emphasizing their potential in contributing to sustainable development. Additionally, research by Zaid, Jaaron, and Bon (2018) [8] demonstrated the impact of GHRM and GSCM practices on sustainable performance. Furthermore, the mediating role of GSCM in enhancing business performance and environmental sustainability has been highlighted in the context of developing countries, which is pertinent to Jordan's pursuit of sustainable development [9].

GHRM and GSCM stand out as two widely discussed topics that are linked to the SDGs [7–10]. A notable gap exists in the integration of GHRM and GSCM; however, research indicates that both can contribute a positive impact to the triple bottom line (TBL) of sustainability performance [8]. GSCM practices can improve business performance and environmental sustainability [9]. Overall, the integration of GHRM and GSCM can contribute to achieving the SDGs, which include legal frameworks for environmental protection and labor rights. The international laws and regulations can provide a legal framework for countries to adopt policies and regulations that align with the SDGs, promote partnerships, and provide tools to achieve SDGs. Hence, it can be inferred that the legal framework plays a crucial role in the execution of Green Human Resource Management (GHRM) and Green Supply Chain Management (GSCM) practices.

However, a notable gap exists in integrating these elements, particularly in Jordan. While there is recognition of the positive impact of integrating GHRM and GSCM on sustainability performance, research indicates a delay in their integration due to gaps in understanding the relationship between GHRM, GSCM, and regulation and legislation. This article aims to address this gap by emphasizing the implications of integrating these elements for organizational sustainability in Jordan. [7,11–13]. In Jordan, the industrial sector constitutes a key fundamental aspect of the Jordanian economy, contributing 22.4% of gross domestic product (GDP) and providing employment for 14.2% of the total workforce, comprising 227,000 jobs, primarily low to medium-skilled positions as of 2021. Therefore, it is important to monitor the response of the industrial sector to government policies promoting sustainability.

## 2. Aim

This study aims to enhance the theoretical framework and address the literature gap by analyzing existing research articles related to GHRM, GSCM, and regulation and legislation, and their connection to the United Nations Sustainable Development Goals (SDGs) in Jordan. From a theoretical perspective, GHRM, GSCM, and regulation and legislation research is still in its infant stage and needs more exploration and development. Although scholars in each of these fields are advancing the roles of Green Supply Chain Management (GSCM), GHRM, and regulation and legislation, there has been a notable delay in the integration of these three contemporary subjects, particularly due to a significant gap in the integration of GHRM, GSCM, and regulation and legislation. Thus, this article emphasizes the implications of GHRM, GSCM, and regulation and legislation integration for scholars, managers, and practitioners within the realm of organizational sustainability in Jordan. Furthermore, the framework for GHRM, GSCM, and regulation and legislation can give a solution for the barriers to implementation.

This aim can be achieved through the following objectives:

1. Critically analyze a review of the relevant literature.
2. Propose a model to fit Jordanian requirements.
3. Present the conclusion and recommendations for further research work.
4. Identify and examine the motivators and barriers.

Therefore, the central aim of this study is to answer the following research question: "How can GHRM, GSCM, and Regulation and Legislation in Jordan be developed to achieve SDGs?". This study also aims to answer the following research sub-questions:

1. How do GHRM and GSCM practices contribute to or hinder the achievement of SDGs in Jordan?
2. What is the existing regulatory framework in Jordan concerning sustainability and how does it impact businesses in terms of GHRM and SCM?
3. What are the main obstacles faced by organizations in Jordan when attempting to integrate GHRM and GSCM practices?
4. How do regulatory constraints pose challenges to the effective implementation of sustainable practices?
5. How do existing regulations and legislation in Jordan support or hinder the adoption of GHRM and GSCM?
6. To what extent can the integration GHRM and GSCM contribute to the achievement of specific SDGs in Jordan?
7. Are there discernible economic, social, and environmental benefits for organizations aligning their practices with SDGs?
8. What are the barriers and key drivers of implementing GHRM and GSCM in Jordan?

## 3. Research Methodology

The methodology approach used in this study involved conducting a review of the relevant literature and accessing databases and search engines to gather information. The methodology outlines a structured approach to gathering and reviewing the relevant literature, conducting database and reference searches using specific keywords and utilizing various sources to collect information. The methodology involves a comprehensive search for articles, reports, and other relevant content from both Arabic and foreign sources, including scientific references, the current literature, theses, formal Jordan reports, internet sites, and electronic libraries. The researchers also utilized search engines and academic references to gather information. See Figure 1 for the PRISMA diagram.

The first step was to conduct a database search using specific keywords related to government regulations and legislations, Green HRM (GHRM), and Green Supply Chain Management (GSCM) in Jordan. The same keywords were used for a reference search in the second step. The chosen keywords allowed the researchers to group the selected papers into three categories: government regulations and legislations, GHRM, and GSCM.

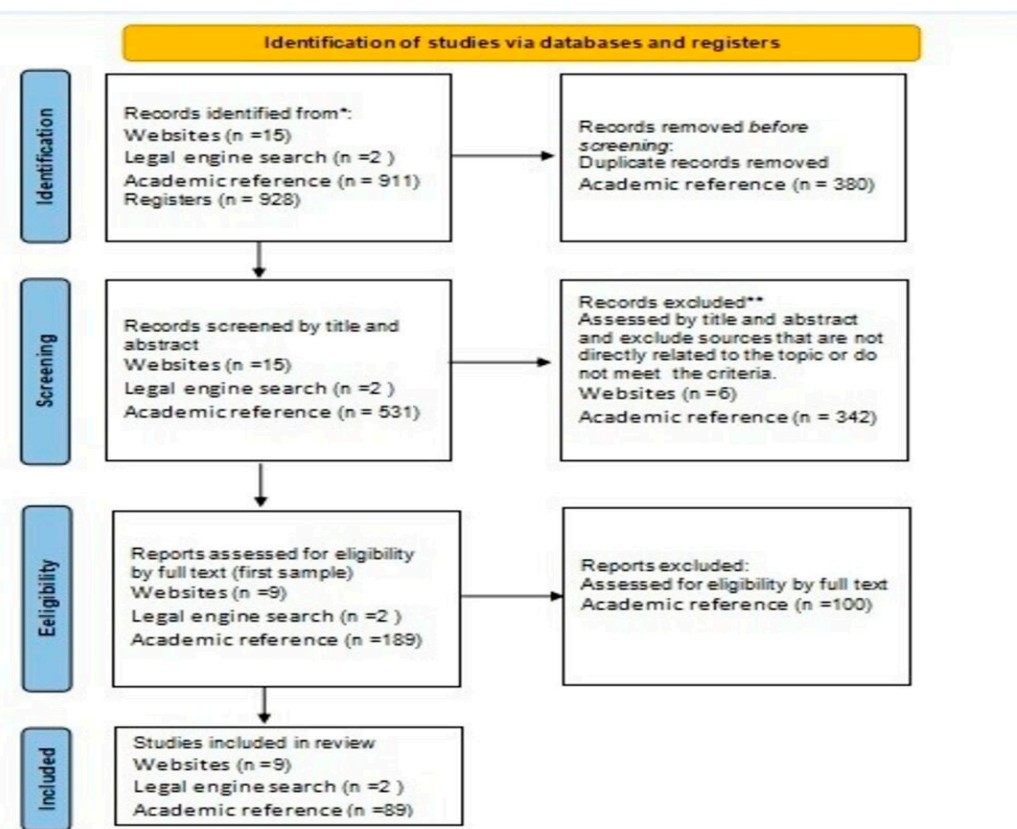

**Figure 1.** PRISMA diagram that Illustrates the progression of studies, starting from retrieval from electronic databases and other sources, through to the selection of studies deemed suitable for inclusion in the systematic review.

For the search on GHRM, keywords such as "sustainable HRM," "green HR," "GHRM practices," "sustainability and HR," "SDGs and GHRM," "SDGs and GHRM practices," "GHRM and GSCM," and "Pillars of sustainability and GHRM" were used. Similarly, keywords like "sustainable SCM," "green SCM," "GSCM practices," "sustainability and SC," "SDGs and GSCM," "SDGs and GSCM practices," "GSCM and GHRM," and "Pillars of sustainability and GSCM" were used for the search on GSCM.

Additional keywords such as "Regulation and SDGs," "regulations and legislations and SDGs," "Regulation and GHRM," "Regulation and GSCM," "regulations and legislations and GHRM," and "Government regulations and legislations and GSCM" were used to cover laws and judicial decisions. Qistas and Qarark search engines, which provide access to original Jordan content including laws and judicial decisions, were also utilized.

These keywords facilitate a comprehensive exploration of the interplay between Green Human Resource Management (GHRM), Green Supply Chain Management (GSCM), regulation, legislation, and their collective influence on promoting Sustainable Development Goals (SDGs). They encompass theoretical, practical, and regulatory perspectives, providing a holistic comprehension of the subject matter.

**Green Human Resource Management (GHRM):**

1. "Sustainable HRM" and "Green HR": these terms directly link HR practices with sustainability goals, emphasizing environmentally conscious human resource management.
2. "GHRM practices": this specifies a focus on the practical aspects of implementing green and sustainable HR policies.
3. "Sustainability and HR": this highlights the integration of sustainability principles into HR functions.

4. "SDGs and GHRM," and "SDGs and GHRM practices": these establish the link between GHRM practices and their contribution to achieving SDGs.
5. "GHRM and GSCM": this recognizes the interconnectedness between Green HRM and Green Supply Chain Management.

**Green Supply Chain Management (GSCM):**

1. "Sustainable SCM" and "Green SCM": these highlight the environmentally friendly aspects of supply chain management.
2. "GSCM practices": this focuses on practical sustainability strategies within supply chain management.
3. "Sustainability and SC": this connects sustainability principles with broader supply chain management.
4. "SDGs and GSCM," and "SDGs and GSCM practices": these demonstrate the alignment between GSCM practices and SDGs.
5. "GSCM and GHRM": this acknowledges the integration of Green Supply Chain Management with Green Human Resource Management.

**Regulation and Legislation:**

1. "Regulation and SDGs," and "Regulations and legislations and SDGs": these explore how regulations and legislations impact or contribute to the achievement of SDGs.
2. "Regulation and GHRM," and "Regulation and GSCM": these investigate the influence of regulations on Green HRM and Green Supply Chain Management.
3. "Regulations and legislations and GHRM," and "Government regulations and legislations and GSCM": this explore the regulatory landscape and its impact on Green HRM and Green Supply Chain Management practices.

Data were collected from Arabic and foreign sources relevant to the research topic, including scientific references, the current literature such as studies, research articles, scientific periodicals, and theses, formal Jordan reports, internet sites, and electronic libraries. The initial sample included 200 articles, which were subsequently narrowed down based on reading the abstracts, discussions, and conclusions. A literature review of 20 articles was conducted, and 80 articles were reviewed to explain the conceptualization of the paper variables. Articles that did not directly relate to the subject issues or that used outdated references were excluded. A summary of the design and methodology of the current research is provided in Figure 2.

In the review and assessment process, the journal articles under consideration were published between 2016 and 2023. The selection of the years 2016 and 2023 is significant due to key developments in Jordan's commitment to sustainable development during this period:

1. The first and second Voluntary National Review (VNR): The VNR provided a comprehensive overview of the country's progress and challenges in implementing the SDGs. This signified Jordan's dedication to transparency and accountability in addressing global sustainability targets.
2. The 2023 agenda for sustainable development including 17 SDGs: the UN adopted SDGs in 2015, and in 2016, the implementation of these SDGs began.
3. Paris Agreement entry into force: The entry into force of the Paris Agreement on climate change in 2016 is another crucial factor. The Paris Agreement, adopted in 2015, represents a global commitment to combat climate change.
4. The choice of 2023 is forward-looking, allowing for an examination of Jordan's continued efforts and advancements in sustainable development. By selecting this year, it provides an opportunity to explore updates on the implementation of sustainable practices, achievements, and challenges faced by Jordan in the years following the initial VNR, the entry into force of the Paris Agreement, and the implementation of SDGs.

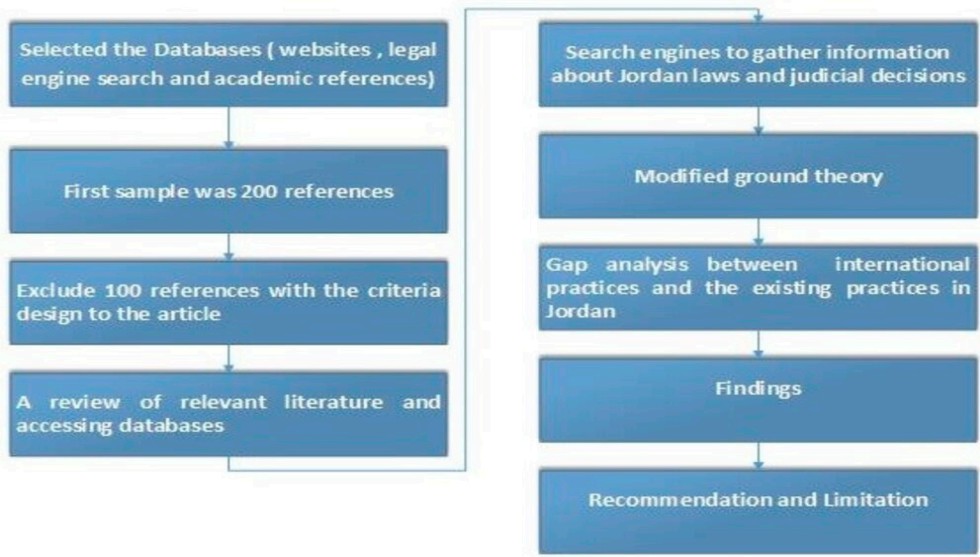

**Figure 2.** The design and methodology of the current research.

In summary, the rationale behind choosing 2016 and 2023 is to capture a pivotal moment when Jordan actively engaged with the global community on sustainable development through the VNR and the Paris Agreement. The subsequent inclusion of 2023 allows for an assessment of the ongoing efforts and progress made by Jordan in the years following these critical milestones.

The search was conducted in the English language, and the sources from online databases were specifically related to the practical side of sustainability. The researchers used various sources to collect information and data from, including the following:

1. Sites Sources:

    1.1 Ministry of Environment website;
    1.2 Ministry of Planning and International Cooperation website;
    1.3 Jordan Chamber of Industry;
    1.4 United Nations websites;
    1.5 USAID/Jordan website;
    1.6 UNEP website;
    1.7 Jordan vision website;
    1.8 Arabsdg unescwa website.

2. Qistas and Qarark search engines.

3. Academic references: From the search, we obtained an initial sample of 200 articles form different databases such as Google Scholar and Scopus. Subsequently, after reading the abstracts, discussions, and conclusions, we conducted a literature review of 86 articles and excluded the remaining articles because they did not directly relate to the subject issues under consideration. A conceptual framework of the research model including the correlations between government regulations and legislations, GHRM, GSCM, and SDGs is shown in Figure 3.

Overall, the theoretical framework presented here sets the stage for understanding the complex interplay between GHRM, GSCM, and regulation and legislation, and their collective influence on achieving SDGs in Jordan. Through a structured methodology, this study seeks to contribute to the advancement of knowledge in this field and provide practical insights for stakeholders in Jordan's sustainability journey.

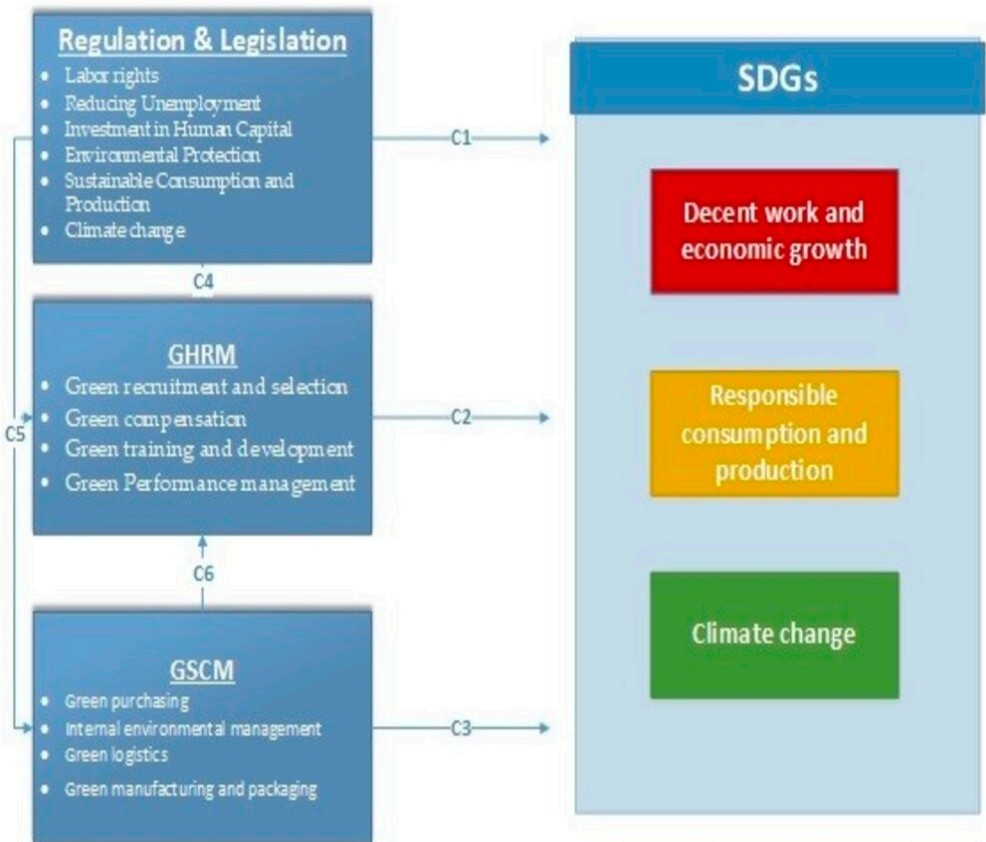

**Figure 3.** A conceptual framework of the research model.

## 4. Literature Review

The literature review encompasses studies on GHRM, GSCM, and government regulations and legislation, emphasizing their individual and collective impacts on sustainable development. Furthermore, this review highlights the specific implications of these factors in Jordan's context and their alignment with SDGs 8, 12, and 13. The literature review is divided into two parts. The first part focuses on the correlations between dependent and independent factors. The second part analyzes the main practices for each factor and the main benefits and barriers.

### 4.1. Correlations

The literature review focuses on exploring the intersection of regulations, legislations, GHRM, GSCM, and SDGs. The second Voluntary National Review report of Jordan in 2022 is used as a primary source for regulations, legislations, and SDGs [4,5]. Brief summaries of the previous relevant literature are mentioned in Table 1.

**Table 1.** Literature review.

| Reference | Title | Methodology | Findings | Gap and Recommendation |
|---|---|---|---|---|
| [14] | SCM and the United Nations Sustainable Development Goals | A study of the existing literature published in the Scopus database between 1991 and 2020 using bibliometric analysis. | A clear connection between the United Nations' SDGs and SSCM was established and how this linkage can bring tangible benefits to organizations was elucidated. | The scope of the research can be expanded to encompass a larger scale to delve into the relationship between SDGs and societal happiness. |
| [15] | GHRM as a strategic tool for fostering the sustainable development of enterprises | A survey was conducted within a randomly selected, representative population sample of 150 emerging enterprises. | A strong positive correlation exists between assessing the impact of individual activities within GHRM on sustainable company development and their effective practical implementation. | This study can raise awareness and disseminate knowledge about the potential impact of Green HRM on promoting sustainable development within organizations. |

**Table 1.** *Cont.*

| Reference | Title | Methodology | Findings | Gap and Recommendation |
|---|---|---|---|---|
| [16] | The Influence of GHRM Practices and Person-Organization Fit: Examining the Moderating Role of the Personal Environmental Commitment | A quantitative and hypothetical–deductive approach was used, and a sample of 204 Portuguese employees responded to an online questionnaire. | Once can employ a triple bottom line approach, the SDGs 13 + 8, and a five-factor measure to assess perceived GHRM practices, encompassing green recruitment and onboarding, green training, green performance management and rewards, green internal communication, and the cultivation of a green sustainable culture. | The impact of all GHRM practices on the person–organization fit is more pronounced among participants with high environmental commitment compared to those with low environmental commitment. |
| [17] | Green recruitment in facilitating the adoption of GSCM practices | A qualitative approach Including 12 in-depth semi-structured interviews across four case companies. | Green recruitment can serve as a catalyst for the adoption of GSCM practices. | Studies should be conducted in the context of other countries. |
| [18] | GSCM Practices in Developing Countries | A single case study named Al-Quds paints, involving the collection of data through semi-structured interviews with both the general manager and the environmental manger. | Manufacturers in Jordan are expressing interest and dedication toward protecting the environment, even in the absence of governmental regulations, by adopting a range of GSCM practices. | One can use multiple cases from various Jordanian manufacturers to examine and illustrate the adoption of Green Supply Chain Management (GSCM) practices. |
| [19] | Model for GSC Adoption: An Empirical Analysis | A quantitative study involving 405 respondents. Qualitative data were obtained through interviews in the industrial sectors in MENA developing Countries. | There exists a significant correlation between the environmental, organizational, and technological dimensions of the practices of firms and Supply Chain (SC) practices. | Government regulations and requirements are important motivators to compel industries to adopt GSC programs and health and environmental programs in the industry. |
| [20] | Green Purchasing: Past, Present and Future | The study analyzed 142 studies from 61 journals published between 1998 and 20. | The study identified the mechanisms of persuasion that motivate consumers to buy green products and provided a clear picture of the contribution of green purchasing to achieving sustainability. | Future studies may take these findings to help develop future research on government regulations and government initiatives. |
| [21] | Adoption level of green practices and its effect on employee' performance technological, organizational and environmental factors | Quantitative research including 2000 employees working in road transport of goods in Turkey. | The technological factors and organizational factors (quality HR) have statistically significant effects on task performance. | The findings can be applied in businesses of different sectors. |
| [22] | Exploring the role of Food Supply Chain Stakeholders in attaining the United Nations Sustainable Development Goals (SDGs) | A literature review using the Delphi method. | Four Sustainable Development Goals (SDGs) have been identified as most important, namely: SDG6, SDG7, SDG12, and SDG13. | Continued efforts are needed to pave the way for achieving the targets of the UN SDGs and exceeding the expectation of all stakeholders in the meat supply chain. |
| [23] | GHRM, coupled with environmental management practices and perceived organizational support, plays a pivotal role in influence organizational citizenship behavior towards the environment | A quantitative study of 117 construction employees in Sungai Petani, Kedah, in Malaysia. | Organizational citizenship behavior toward the environment among employees is unaffected by these factors. It is probably attributed to a deficiency in the implementation and execution of environmentally friendly practices, coupled with a lack of awareness among the employees regarding the importance of environmental protection. | It is advisable for stakeholders (employees, management, organization, industry, and government) to revisit the existing policy and enforce the rules and regulations more rigorously to all industries and all among employees at all levels. |
| [24] | Assessing the Environmental Performance of Indonesian Small and Medium-sized Enterprises (SMEs) influenced by Green Supply Chain (GSC) Practices with Moderating Role of Green Human Resource (GHR) Practices | The data were collected in a questionnaire from SMEs of Indonesia. | Green Supply Chain (GSC) practices have a significant influence on the environmental performance of firms; green purchasing and logistics exert influence on the environmental performance of firms. The study revealed that moderating the role of GHRM was not evident between green purchasing and environmental performance, and GHR practices did moderate the relationship between green logistics and environmental performance. | Take a proactive approach to encourage the adoption of green small and medium-sized company (SMC) practices, ensuring that the employees are environmentally conscious and adopt Green Human Resource (HR) practices. |

**Table 1.** *Cont.*

| Reference | Title | Methodology | Findings | Gap and Recommendation |
|---|---|---|---|---|
| [25] | Factors affecting managers' intention to adopt GSCM practices | Quantitative research involving 376 manufacturing firms in Jordan. | Supplier's commitment, environmental sustainability, customer satisfaction, and cost factors are the most significant drivers. | This study employed a convenient sampling approach, thereby limiting the generalizing capability of its findings. |
| [26] | Factor affecting sustainability integration | Qualitative content analysis of public construction industry in Jordan. | The factors encompass the current process of public procurement and contract development, the existing regulations and government support, and the expertise and knowledge of professionals. | One can discover the best practices and programs linked with integrating sustainability in the public construction industry, while also identifying barriers that obstruct the realization of sustainability. |
| [27] | Barriers for Integrating Sustainability into Public Works | Qualitative research. | The findings revealed the importance of the existing process of public procurement and contract development, the absence of regulations and government support, and a shortage of professionals with expertise and knowledge. | Recommendations for future work include validating these findings through quantitative methods. Additionally, further solutions to overcome these identified barriers need to be investigated. |
| [28] | Antecedents for greening the workforce: Implications for GHRM | A literature review. | The findings showed that essential factors influencing the implementation of Green Human Resource Management (GHRM) practices encompass the 'green selection facility', 'green recruiting facility', 'green organizational culture', 'green purchasing, 'green strategy toward ES', 'regulatory forces towards ES', and 'top management commitment toward greening the workforce'. | In future research endeavors, studies can explore additional regions to broaden the scope of the study. |
| [29] | When knowledge management matters: interplay between green human resources and eco-efficiency | A quantitative study with a sample of 178 employees of financial banks in Brazil. | Regulatory forces toward eco-efficiency, green pay and reward, and green facilities will benefit sustainable development. | Enhancing the effectiveness of the studied bank's eco-efficiency program could be achieved by implementing a green team. |
| [30] | Highlighting the importance of Sustainable Human Resource Management in facilitating the adoption of Sustainable Development Goals | A systematic literature review. | Organizational change is underlain by motivations such as competitiveness, legitimacy, and ecological responsibility. | Research gaps in the current literature have been identified. Suggested avenues for further research in the field of sustainable management have been proposed. |
| [31] | GHRM: Best practice of Attaining Sustainable Development Goals | Descriptive research in India. | Green Human Resource practices are highly beneficial for achieving the Sustainable Development Goals at the micro-level within industries. | If every industry would introduce Green Human Resources (GHR), there would be the potential to significantly contribute to macro-level sustainable development. |
| [32] | Establishing a connection between green HR Practices and Environmental Economics Performance: Exploring the Role of Green Economic Organizational Culture and Green Psychological Climate | A quantitative study approach. | GHRM exerts a positive influence on ecological factors, including green psychological climate, green organizational culture, and sustainable environmental efficiency. Moreover, green organizational culture and green psychological climate positively mediate the relationship between GHRMP and SEF. | Adopting GHRM strategies and enhancing technical innovations can contribute to improving sustainability and economic performance. |
| [33] | Barriers to GSCM: An emerging economy context | A questionnaire survey given to Bangladeshi textile professionals in the operations and supply chain management division. | One of most important barrier is the lack of government regulations. | Informing the relevant policymakers about the barriers prevailing in emerging economies that hinder the adoption of Green Supply Chain Management (GSCM) practices. |

Various scholarly works by Cesário et al. (2022); Sachin and Aradhana (2019); Nour Chams and Josep García-Blandón (2019); and Bombiak and Marciniuk-Kluska (2018) discuss the relationship between GHRM and SDGs [15,16,30,31]. Similarly, works by Srivastava et al. (2022); Djekic et al. (2021); Hazaea et al. (2022) explore the intersection of GSCM and SDGs [14,20,22]. Furthermore, this review includes studies by Chams and García-Blandón (2019); İktisadi Ve et al. (2020); Moraes et al. (2018); Almahera (2018); Md et al. (2019) that delve into the connection between regulations, legislations, and GHRM [21,23,28–30]. Similarly, research by Moktadir, et al. (2019); Hanna (2021); Tumpa et al. (2019); Alnsour (2019); Alnsour (2020); Abdellatif and Graham

(2019); Jum'a et al. (2021) investigates the relationship between regulations, legislations, and GSCM [18,19,25–28,33]. Finally, this review also covers studies by Alnsour (2019); Alnsour (2020); Jermsittiparsert et al. (2019); Abdellatif (2021); Chavez et al. (2020); Md et al. (2019); Chiappetta Jabbour and Sousa Jabbour (2016) that discuss the intersection of GHRM and GSCM [7,12,17,24,26,27].

*4.2. Analysis*

4.2.1. Sustainable Development Goals

In 2015, the United Nations developed the Sustainable Development Goals (SDGs) to address three pillars that build upon the progress and the agenda set by the Millennium Development Goals in 2000. Ratified in January 2016 following the NY summit, the SDGs delineate specific responsibilities for developed nations, emphasizing the importance of prioritizing sustainability and partnership in communication with developing countries [2,5].

The concept of the "three pillars" within companies and the triple bottom line (TBL) approach (profit, planet, and people) are fundamental to many institutions, and government agencies today. SDGs aggregate according to the social, economic, and environmental pillars through the direct impacts and policy goals of the SDG pillars [1]. For example, SDG8 and SDG9 are closely linked to sectors that produce jobs, decrease poverty, and improve people's lives. SDG7, SDG 12, and SDG 13 are closely linked to manufacturing sectors that are related to SCM. Table 2 shows the classification of the SDGs [3,14].

**Table 2.** Classification of SDGs.

| Pillars | SDGs |
| --- | --- |
| Social | SDG1, SDG2, SDG3, SDG4, SDG5, SDG7, SDG11, and SDG16 |
| Economic | SDG8, SDG9, SDG10, and SDG12 |
| Environmental | SDG6, SDG13, SDG14, and SDG15 |

As shown in the second Voluntary National Review report of Jordan, notable progress in terms of indicators was made toward SDGs 9 and 12, while the least progress was made toward SDG8. A partnership with the private sector, initiated in 2015, played a crucial role in implementing many projects aligned with SDG7. These projects focused on aspects such as the disposal and transport of dangerous and medical waste, as well as the training of cadres in the health sector, aligning with the objectives outlined in SDG12 [4,5]. Table 3 shows the three SDGs and how they achieve green growth.

**Table 3.** Goals of Green Growth National Action Plan of Jordan linked to SDGs 8, 12, and 13.

| Goals of Green Growth National Action Plan | SDGs |
| --- | --- |
| Sustainable Economic Growth | Decent work and economic growth |
| Resource Efficiency | Responsible consumption and production |
| Climate Change Adaptation and Mitigation | Climate change |

Overall, GHRM, GSCM, and government regulations and legislations can help organizations achieve their sustainability goals while promoting economic growth, environmental protection, and social development, which are the main objectives of the UN SDGs.

4.2.2. Green Human Resource Management

Green HRM is a concept that aims to motivate employees to help organizations achieve their environmentally friendly goals and contribute to environmental sustainability by showing environmentally friendly behavior. Several companies have successfully implemented GHRM practices [12,13]. For example, Patagonia has implemented GHRM

practices to promote environmental sustainability, including offering employees environmental training and encouraging them to participate in environmental initiatives [13]. IBM has implemented GHRM practices to promote environmental sustainability, including offering employees incentives to reduce their carbon footprint and promoting sustainable transportation options [34]. Other companies that have successfully implemented GHRM practices include Unilever, Interface, and Timberland [12,13].

The pro-environmental HR practices identified in the research contribute to the sustainable development of enterprises by incorporating environmental practices into the area of human resource policy. This leads to a reduction in the negative impact of business operations on the environment, which is an essential aspect of sustainable development. By adopting these practices, companies can improve their environmental performance, reduce costs associated with waste management and energy consumption, and enhance their reputation as socially responsible organizations [15].

There is some variation in the literature regarding the three pillars of green HRM. However, based on the provided search results, the three pillars of green HRM can be summarized as three overarching factors. The first pillar is environmental sustainability and it involves promoting ecological responsiveness and the sustainable development of resources while involving employees' commitment and engagement toward the organization's goals and practices. It includes practices such as green recruitment, selection, hiring, learning, training, and development, performance management and appraisals, total reward systems, and other practices that promote sustainable workplace practices [35–37]. Secondly, social sustainability is a pillar that involves developing employees' skills, knowledge, and attitudes about environmental conservation and environmental management initiatives through training and development programs. It also includes promoting pro-environmental and pro-social corporate management methods and practices [38,39]. The third pillar is economic sustainability and this pillar involves promoting sustainable economic development that eradicates the depletion of natural resources. It includes practices such as green appraisals, recruitment, training, and development that substantially contribute to the sustainability of organizations [37].

Despite its increase in popularity, there is a lack of consensus on Green Human Resource Management (GHRM) practices [24]. Several authors have developed different approaches to categorize GHRM practices. For example, İktisadi Ve et al. [21] identified four GHRM practices: green employee acquisition, green employee appraisal, and green employee rewards; Luay Jum'a [25] proposed a unidimensional measure for GHRM; Alnsour, Moawiah [26] examined green hiring, green training and involvement, and green performance management and compensation; Kara, Karahan, and Edinsel, Sercan [10] studied green recruitment, green rewards and compensation, and environmental training; Tang et al. Alnsour, Moawiah [27] proposed five GHRM practices: green recruitment and selection, green training, green performance management, green pay and reward, and green involvement. Nawangsari and Sutawidjaya [28] suggested the following practices: green recruitment, green performance management, green training and development, green compensation, and green employee relations. Moraes et al. [29] proposed a model comprising seven GHRM practices: green compensation management, green health and safety, green job design, green labor relations, green performance management, green recruitment and selection, and green training and development. According to the article by Rawashdeh [40], the GHRM practices that were studied in Jordanian health service organizations include green recruitment and selection, green training and development, and green rewards.

Green Human Resource Management (GHRM) can contribute to the attainment of various Sustainable Development Goals (SDGs). To elucidate how organizations can convert Human Resource Management (HRM) practices into "green" initiatives that bolster and support corporate sustainability, the distinct dimensions or practices of GHRM are elaborated on below and shown in Figure 3.

Green Recruitment and Selection

Green recruitment involves the process of selecting and hiring individuals with environmental management skills, mindsets, and behaviors that promote sustainability objectives. It can be linked to the SDGs 8, 12, 13, and 15 by promoting sustainable workplace practices and reducing their operations' environmental impact. For instance, this can include the use of automated application processes, green interview processes through Skype and Zoom, the advertisement of jobs on e-career portals, psychometric tests to evaluate applicants' green behavior value system, digital tools, telecommuting and remote work, and the inclusion of green competencies in job requirements and advertisements [31,35,36,41,42].

Green Compensation Management (Reward Management)

This refers to the incorporation of environmental management into compensation and reward systems, including financial and non-financial rewards. It can be linked to the SDGs 3, 7, 8, 12, and 13 by promoting sustainable workplace practices and reducing their operations' environmental impact. For example, this can include executive stock option compensation oriented around a 3-year or more vesting period, mapping a Green Compensation Package (GCP) with employee rights, and using green performance management and compensation to mediate the relationship between green hiring and sustainable performance [34–36,42–44].

Green Training and Development

Green training and development is a combination of coordinated activities that encourage and empower employees to acquire skills related to environmental protection, with a focus on addressing essential environmental issues that play a pivotal role in achieving environmental objectives. It can be linked to the SDGs 4, 8, 12, 13, and 17. ISO 14001 states that employees at all levels of the organization should understand the environmental system and how to effectively have a positive impact on the environment. Here are some ways in which green training can contribute to sustainable workplace practices: developing green abilities, promoting environmental awareness, improving employee commitment toward the environment, and incorporating green training into organizational policies [31,34–37,41,42,45,46].

Green Performance Management

This process encourages employees to improve their professional skills while also considering the environmental concerns and policies of the company. It includes indicators for evaluating green behavior, such as compliance with standards, progress in the acquisition of environmental responsibilities, and corporate-wide environmental performance standards. It can be linked to the SDGs 7, 8, 12, 13, and 15 [31,35,36,42,47].

Based on the search results, there are limited specific industries in Jordan that have implemented Green HRM practices. However, some studies have identified some industries that have implemented GHRM practices in Jordan. These industries include health service organizations [40] and five-star hotels [48]. These practices include green recruitment and selection [40], online interviews and training, green payroll, electronic signatures to avoid wasting paper, and using porcelain cups for tea [48].

Jordanian companies can assess their current level of GHRM adoption by considering various factors such as green recruitment and selection, total reward systems, green training and development, and green performance management. Secondly, they can evaluate the level of awareness among employees and assess their knowledge and expertise to identify areas for improvement. Also, Jordanian companies can use technology to enable and facilitate GHRM practices, such as digital tools for recruitment and employee communication, telecommuting and remote work, and electronic documentation. Jordanian companies should evaluate the extent to which their stakeholders, including customers, employees, and investors, are demanding environmentally responsible practices and policies. Furthermore, they should assess the benefits of adopting green HRM practices, such as cost

savings, improved reputation, and increased employee engagement. Finally, top management commitment is crucial for organizations' sustainable development. By evaluating these factors, companies can identify areas for improvement in their adoption of green HRM practices and develop strategies to promote sustainable workplace practices and reduce their environmental impact [49,50].

Based on the provided information, Table 4 summarizes the examples of Green HRM practices.

**Table 4.** Examples of Green HRM practices.

| Practice | Example | References |
|---|---|---|
| Green Recruitment and Selection | The use of automated application processes, green interview processes through Skype and Zoom, the advertisement of jobs on e-career portals, and the inclusion of green competencies. | [31,35,36,41,42] |
| Green Compensation Management (Reward Management) | Executive stock option compensation oriented around a 3-year or more vesting period, and mapping a Green Compensation Package (GCP) with employee rights. | [34–36,42–44] |
| Green Training and Development | Providing training to employees on environmental issues, and incorporating green training into organizational policies. | [31,34–37,41,42,45,46] |
| Green Performance Management | Evaluating green behavior, compliance with standards, progress in environmental responsibilities, and corporate-wide environmental performance. | [31,35,36,42,47] |
| Green HRM Industry Examples | Health service organizations and five-star hotels. | [40,48] |

### 4.2.3. Green Supply Chain Management

Green Supply Chain Management (GSCM) is an approach that integrates environmental, social, and economic issues. Several companies have successfully implemented GSCM practices [51–55]. For example, Walmart has implemented GSCM practices to reduce waste, improve energy efficiency, and promote sustainable sourcing [51]. Nike has implemented GSCM practices to reduce carbon emissions, improve labor conditions, and promote sustainable material sourcing [52]. Toyota has implemented GSCM practices to reduce waste, improve energy efficiency, and promote sustainable material sourcing [53]. Other companies that have successfully implemented GSCM practices include Coca-Cola, Dell, and Unilever [54,55].

GSCM can play a significant role in achieving the SDGs by helping companies attain the three pillars of sustainability: environmental, social, and economic. The goal of GSCM is to balance economic and environmental performance to stay competitive while conforming with regulatory and community pressures. Here are some examples of social, environmental, and economic practices in GSCM. First of all, social practices include fair labor by ensuring that workers are treated fairly and ethically, community involvement by engaging with local communities to understand their needs and concerns, and diversity and inclusion through promoting diversity and inclusion in the workplace. Secondly, environmental practices include green purchasing by buying environmentally friendly products and materials, green manufacturing by using eco-friendly production processes and technologies, green distribution such as reducing carbon emissions by optimizing transportation routes and modes, reverse logistics, green packaging by using recyclable or biodegradable packaging materials, environmental education and awareness such as by providing training to employees on environmental issues, internal environmental management by implementing systems to monitor and reduce environmental impacts, and saving resources and reducing waste by recycling or reusing waste materials. Finally, economic practices include sustainable supply management by working with suppliers to improve their sustainability performance, cost reduction through eco-efficiency measures such as

waste reduction, energy efficiency. and using waste materials to create new products, and risk management by identifying risks associated with environmental, social, and economic issues in the supply chain and managing them effectively [56–61].

According to recent research, there are several dimensions of GSCM practices that have been identified. These dimensions include green purchasing, manufacturing, distribution, packaging, and warehousing, environmental education, internal environmental management, and investment recovery. Additionally, a conceptual framework proposed by Slašťanová et al. [61] is organized into 3 environmental dimensions, 21 categories, and 64 green practices. The green practices that are mentioned in the research also includes practices related to GHRM that include green performance such as environmental audit programs, monitoring, environmental accidents and benchmarking, and support and education. GSCM assure the effectiveness of public and company policies in greening their operations, increasing the market share, improving the company image and reputation, and increasing profits.

GSCM is a growing area of interest in Jordan, particularly in the manufacturing [18], renewable energy [62], pharmaceutical [63,64] and tourism sectors [65]. By adopting GSCM practices, companies can contribute to the achievement of many SDGs. The different dimensions of GSCM are discussed the following sections.

Green Purchasing

One of the critical dimensions of Green Supply Chain Management involves actively seeking specialized suppliers who provide environmentally friendly and sustainably produced materials in accordance with environmental standards. This activity considers environmental impacts in addition to factors such as price and quality when making comparisons. It also emphasizes the damaged or lost materials whenever possible. The selection of vendors who have the ISO 1400 certificate is expected to eliminate environmental risk. The reduction in environmental risk improves profitability and then will have those vendors for long-term businesses [66]. Green purchasing behavior can contribute to several SDGs, such as the SDGs 7, 8, 9, 12, 13, 14, and 15 [20,65,67–73]. This can include specifications for suppliers, green packaging, supplier selection, supplier audits, and evaluations of second-level suppliers [61].

Internal Environmental Management

This is an encompassing practice that includes senior management commitment, support from mid-level management, cross-functional collaboration, the integration of total quality management, the implementation of an environmental auditing program, and ISO 14001 certification. ISO 14001 is an international standard for environmental management systems, providing organizations with a framework to effectively manage their environmental impacts and improve their overall environmental performance. Many companies use ISO 14001 as a tool to implement and maintain effective environmental management practices [61,66]. IEM can contribute to several SDGs, such as the SDGs 8, 9, 12, 13, 15, and 17 [74–77].

Green Logistics

Logistics activities cover transportation and warehousing, as well as all activities related to the selection of the best transportation means, load carriers, transportation routes, and green storage to reduce the environmental impact of the whole supply chain [65,66]. Green logistics can contribute to several SDGs, such as the SDGs 8, 9, 11, 12, 13, 14, and 17 [68,78–83]. An example of green logistics is reverse logistics, such as remarketing, taking back packaging for inspection, classification, and warehousing, and green building [61].

Green Manufacturing and Packaging

Green manufacturing and packaging adheres to practices that meet environmental standards, emphasizing efficiency, minimal or zero waste generation, and economic advantages for organizations. Consequently, organizations are now adopting a triple principle encompassing lower costs, recyclability, and reusability as part of a green environmental approach to enhance competitiveness [66]. This approach includes aspects such as the quality of internal service, cleaner production, inventory management plans, internal green production, designing products to avoid or eliminate the use of hazardous substances or their manufacturing processes, designing products for reuse, recycling, material recovery, component part recovery, designing based on resource efficiency, and implementing green distribution practices such as eco-friendly packaging and documentation [61].

Green manufacturing and packaging can contribute to several SDGs, such as the SDGs 1, 7, 8, 9, 12, and 13 [69,82,84–86]. The United Nations Environment Program has outlined two crucial policy priorities for advancing green manufacturing. Firstly, there is a recommendation promoting closed-cycle manufacturing and adopting lifecycle approaches, supported by recovery and recycling infrastructure. Secondly, regulatory reforms are suggested to facilitate improvements in factor efficiency, energy use, and manufacturing settings [85].

Jordanian businesses can benefit from implementing GHRM practices in several ways. These include reducing costs, improving operational efficiency, enhancing brand reputation, complying with regulations, and contributing to environmental sustainability. By adopting sustainable practices in their SC, businesses can also create value for their stakeholders and gain a competitive advantage in the marketplace [61].

Jordanian companies can assess their current level of GSCM adoption by considering the following factors:

1. Barriers to adoption: companies can assess the barriers to GSCM adoption by identifying factors such as a lack of support from the government, cost implications, and a lack of customer awareness, all of which were identified as significant barriers to GSCM adoption in Jordanian firms [18,27,61,87].
2. Key drivers of GSCM adoption: companies can assess the key drivers of GSCM adoption by identifying factors such as government regulations and legislations, stakeholder pressure, global competition, financial factors, and the awareness level of customers, all of which were found to be drivers of GSCM adoption in Jordanian industrial firms [27,88].
3. GSCM practices in the industry: companies can assess their GSCM practices by analyzing their processes in relation to green supplier selection, green purchasing, green production, green design, green distribution, and reverse logistics, all of which were identified as key GSCM practices in Jordanian manufacturing firms [65].
4. Intellectual capital: companies can assess the impact of intellectual capital on GSCM adoption by considering the impact of GSCM dimensions, such as green IT, green manufacturing and packaging, green storing, green purchasing, and green marketing, on the quality of services in renewable energy companies in Jordan [66].
5. ISO 14001: This framework can help your organization reduce waste and pollution, maintain regulatory compliance, and build a presence as an environmentally conscious company. It can also help you raise profits as a recognized "green" company [61,66,87].

Based on the provided information, Table 5 summarizes the examples of GSCM practices.

**Table 5.** Examples of GSCM practices.

| Practice | Example | References |
|---|---|---|
| Green Purchasing | Seeking suppliers of environmentally friendly materials and selecting vendors with ISO 14001 certification | [20,65,67,69,71–73,76] |
| Internal Environmental Management | Senior management commitment, cross-functional collaboration, ad ISO 14001 certification | [61,66] |
| Green Logistics | Optimizing transportation routes and modes, implementing reverse logistics, and using green storage | [68,78–83] |
| Green Manufacturing and Packaging | Producing with minimal waste generation, designing products for reuse and recycling, and using eco-friendly packaging materials | [61,69,82,84–86] |
| GSCM Industry Examples | Manufacturing, renewable energy, pharmaceutical, and tourism sectors | [18,62–65] |

### 4.2.4. Government Regulations and Legislations

Sustainability is a paramount global concern that transcends borders, encompassing a wide range of issues related to economic, social, and environmental well-being. In alignment with the United Nations Sustainable Development Goals (SDGs), Jordan has demonstrated a commitment to addressing these global challenges at the national level. This introduction will delve into the context of sustainability goals 8, 12, and 13 within the framework of Jordanian regulations, shedding light on the country's efforts to promote economic growth, responsible consumption and production, and climate action, while citing relevant references to substantiate these initiatives. In doing so, it becomes evident that Jordan, as a nation, plays a pivotal role in the global pursuit of sustainable development, actively engaging with these goals to secure a better future for its people and the planet. We will now explore how Jordan has integrated three specific SDGs—8, 12, and 13—into its regulatory landscape. In this regard, United Nations goals 8, 12, and 13 hold significant prominence, guiding Jordan's efforts toward advancing economic and social sustainability and addressing environmental concerns [2,4–6].

### Labor Rights

A cornerstone of SDG 8 in Jordan is the safeguarding of labor rights. The government has implemented labor laws and regulations that establish a legal framework for fair wages, working hours, and leave policies, and prevent discrimination in the workplace. These measures are designed to protect workers from exploitation and promote equitable employment practices. The Jordanian Labor Law No. 8 of 1996 is the primary source of legislation that regulates the relationship between employers and employees in the Hashemite Kingdom of Jordan [89,90]. In mid-May 2019, this law was amended by Law No. 14 of 2019, which was published in the official gazette. The main amendments to the law include modifications to various aspects of the employment relationship, including wages, overtime, parental leave, annual leave, childcare, retirement system, and the resolution of wage-related disputes. In addition, employers are committed to the principle of equal pay for work of equal value, and strict penalties are imposed on employers in cases of wage discrimination when work is of equal value.

Note: According to Article 33 of the Jordanian Labor law, Jordan 53 of 1996 as amended in 2019, the employer shall be punished with a fine of no less than five hundred Jordanian dinars and not exceeding one thousand Jordanian dinars for each instance in which they pay an employee a wage less than the minimum wage or discriminate between genders for work of equal value in terms of wages. In addition, the employee shall be entitled to wage differentials, and the penalty shall be doubled for each repeated violation [4,89–92].

In 2017, the Jordanian government adopted the Flexible Working Hours Law to promote women's economic participation; in 2018, the Flexible Work Instructions were issued. These instructions encompass different employment arrangements, including

telework, part-time work, and flexible working hours. These forms of employment options play a vital role in advancing women's economic participation in both the public and private sectors [89,90,93].

Economic Diversification

A diversified economy is vital to sustainable growth. Jordan recognizes this and has been actively working on diversifying its economic sectors, reducing dependency on specific industries, and encouraging innovation and entrepreneurship. This approach aims to create a resilient and dynamic economic landscape capable of withstanding external shocks [4,94].

Jordan can attain economic diversification through a variety of strategies, including fostering high-skill export sectors, boosting tourism, attracting foreign portfolio investments, exploring renewable energy sources, implementing fiscal consolidation, and emphasizing innovation and technological advancement. To achieve these objectives, Jordan may need to establish supportive policies and regulations. For instance, the government could offer incentives to businesses investing in high-skill export sectors, such as tax incentives or subsidies. Additionally, the government could formulate policies that promote the growth of renewable energy, such as implementing feed-in tariffs or net metering. Moreover, streamlining regulations and reducing bureaucracy can enhance Jordan's business environment, potentially attracting more foreign investments [95–97]. For instance, the Jordanian Environmental Investment Law No. 21 of 2022 entered into force with the aim of providing a conducive environment for existing investments and creating attractive conditions for investment, in line with the vision of economic modernization. As a result, Jordan is better positioned to navigate economic challenges, foster job creation, and ensure long-term prosperity for its citizens [4,89,90,95–97].

Reducing Unemployment

A central challenge for Jordan is addressing unemployment, particularly among its youth population. To achieve SDG 8, Jordan has been implementing strategies to boost job creation, enhance vocational training programs, and encourage private sector growth. These initiatives aim to reduce unemployment rates and provide opportunities for all segments of society [89]. An example of this is the Jordanian Environmental Investment Law No. 21 of 2022 which encourages investment in economic zones. The government can provide the necessary infrastructure, tax incentives, and customs facilities to attract investors and establish industrial and commercial projects. These projects can generate employment opportunities for the youth and promote sustainable development in deprived areas. It is also necessary to enhance transparency and combat corruption in the economic and business environment. Policies and laws should be implemented to encourage fair competition and protect workers' rights. In addition, the role of institutions responsible for monitoring the labor market should be strengthened, ensuring a safe and healthy work environment. Strengthening cooperation with the private sector is an effective solution to the unemployment problem in Jordan. The government and the private sector can work together to create new jobs and boost the economy [4,89,90].

Investment in Human Capital

Jordan places a strong emphasis on investing in its human capital. This includes improving education and skills development programs to equip its workforce with the abilities needed to thrive in a competitive global economy. This investment is integral to achieving both economic growth and decent work. Jordan's government and private climate-responsive expenditure and capital financing instructions for the year 2022, issued in accordance with the provisions of Article 9 of Climate Change Law No. 79 of 2019 and published in the official gazette on 2 January 2022, play a crucial role in strengthening human capital. These operations represent a vital means of directing investments toward areas that promote sustainable development and secure the future of Jordanians. By

investing in sustainable and climate-friendly projects, these operations facilitate the creation of new and sustainable job opportunities, contributing to employment opportunities and improving living standards and income potential for citizens [4,89,90].

This financial allocation also aims to develop education and vocational training and to enable the workforce to acquire skills that meet the requirements of sectors responding to climate change. Investing in projects that promote sustainability and improve the environment contributes to improved health and quality of life, which subsequently enhances human capital [4,6].

Moreover, redirecting investments toward new projects and industries in the environmental sector increases the demand for the workforce and expands employment opportunities. Finally, directing investments toward environmentally friendly projects enhances environmental safety and protects natural resources, thus contributing to sustainable development.

Environmental Protection

The Environmental Protection Law No. 6 of 2017 is a law in Jordan that aims to protect the environment and establish the Ministry of Environment as the authority responsible for environmental protection. This law addresses a range of environmental protection facets, including facility permits, harmful substances, and waste management. It is crucial to emphasize that Jordan has additional environmental protection laws in place [91].

In Jordan, Law No. 52 of 2006 is also focused on environmental protection. Furthermore, Environmental Protection Law No. 42 of 2014 strives to protect and maintain the natural balance of the environment and its resources, combat pollution and its consequences, and promote sustainable development. Additionally, Regulation No. 68 of 2020, Hazardous Materials and Waste Management System, was issued in accordance with Articles 6 and 7 of Environmental Protection Law No. 6 of 2017. Nonetheless, currently, Jordan lacks a specific legal framework or a national strategy for the management of solid waste. This absence of a legal framework is leading to the inappropriate disposal of solid waste, giving rise to public health risks, adverse environmental impacts, and socio-economic challenges. In July 2021, a new law was enacted to impose stricter penalties for littering, with a fine of no less than JD 50 and no more than JD 500 for each person that disposes of waste improperly, whether through littering or other means. [90,91].

Sustainable Consumption and Production

The National Action Plan for Sustainable Consumption and Production (NAP-SCP) was developed in collaboration with Jordan's Ministry of Environment, as part of the EU-funded SwitchMed program, with advisory services and technical support provided by the United Nations Environment Program (UNEP). The purpose of this plan is to align Jordan's initiatives with the 2030 Agenda and Sustainable Development Goals (SDGs). The National Action Plan for Sustainable Consumption and Production (NAP-SCP) addresses 12 SDGs in three key sectors: (1) agriculture and food production, (2) transportation, and (3) waste management. This plan was developed in Jordan through nationally recognized processes, involving more than 300 national-level multi-stakeholder participants [5,6,98–100].

Ensuring sustainable consumption and production patterns includes optimizing resource efficiency, sustaining infrastructure, granting access to basic services, creating environmentally friendly employment opportunities, and improving the quality of life for all. Applying sustainable consumption and production patterns also helps achieve comprehensive development plans, reduce future economic, environmental, and social costs, enhance economic competitiveness, and alleviate poverty [4–6,89,90,98,99].

For example, Law No. 8 of 2022 on Public Procurement is a law in Jordan that governs public procurement operations and transactions. This law applies to all government procurement activities and transactions, including the procurement of goods, works, and services. This law aims to ensure transparency, fairness, and competition in government procurement processes. This law also covers various aspects of procurement, including

procurement planning, tendering, evaluation, contract awarding, and contract management, which could potentially include environmental considerations. For example, this law could require that procurement officials consider the environmental impact of goods, works, and services when evaluating bids. Additionally, the law could encourage the procurement of environmentally friendly products and services [89,90]. The Renewable Energy and Energy Efficiency Law No. 13 of 2012 aims to increase the percentage of renewable energy sources in the total energy mix, promote investment in renewable energy, and contribute to environmental protection and sustainable development. The law allows domestic and international companies to bypass a previously complex bidding process and negotiate directly with the Minister of Energy to ease the project implementation process. In 2014, Royal Decree No. 33 amended the Law on Renewable Energy and Energy Conservation, stating that all renewable energy and energy efficiency systems and devices will be exempted from customs duties and sales tax.

Climate Change

Jordan places a significant emphasis on addressing climate change and its ramifications, especially given its commitment to the Paris Climate Agreement. Jordan has actively undertaken numerous projects and initiatives aimed at decreasing greenhouse gas emissions and bolstering its capacity to adapt to anticipated climate shifts [6,98,99]. Jordan recognizes the urgency of addressing climate change and also underscores its dedication to global efforts to curb global warming by referencing the Paris Climate Agreement.

Jordan, despite its modest size and persistent challenges, is steadfastly dedicated to the attainment of UN Sustainable Development Goals 8, 12, and 13. Through robust policies and well-crafted legislation, Jordan unwaveringly champions sustainability across all facets of existence while preserving the environment. This commitment not only serves as a guiding beacon for Jordan's own development but also positions the country as a significant contributor to global sustainability across economic, social, and environmental domains. For example, Jordan's Climate Change Law No. 79 of 2019 was enacted in accordance Article 30 of the Environmental Protection Law No. 6 of 2017. This law aims to set the institutional and regulatory framework on climate change, particularly within the government. This law includes provisions for institutional arrangements, mainly at the national level, for carrying out climate change activities. It also covers emissions and other climate change-related issues. [2,4–6,90]. Table 6 outlines the laws and regulations relevant to various aspects of sustainable development in Jordan, along with brief descriptions of their key provisions, and the associated references.

**Table 6.** Laws and regulations.

| Aspect | Law/Regulation | Key Provisions/Description | References |
|---|---|---|---|
| Labor Rights | Jordanian Labor Law No. 8 of 1996 (amended by Law No. 14 of 2019) | Establishes a legal framework for fair wages, working hours, and leave policies and prevents discrimination. Includes penalties for wage discrimination. | [89–92] |
| | Flexible Working Hours Law (2017) and Instructions (2018) | Promotes women's economic participation through flexible employment arrangements such as telework, part-time work, and flexible working hours. | [89,90,93] |
| Economic Diversification | Jordanian Environmental Investment Law No. 21 of 2022 | Provides a conducive environment for investments, fostering economic modernization, job creation, and long-term prosperity. | [4,89,90,95–97] |

**Table 6.** *Cont.*

| Aspect | Law/Regulation | Key Provisions/Description | References |
|---|---|---|---|
| Reducing Unemployment | Jordanian Environmental Investment Law No. 21 of 2022 | Encourages investment in economic zones to create employment opportunities, with infrastructure, tax incentives, and customs facilities support. | [4,89,90] |
| Investment in Human Capital | Climate-Responsive Expenditure and Capital Financing Instructions for 2022 | Strengthens human capital through education, vocational training, and sustainable projects, contributing to employment and improved living standards. | [4,89,90] |
| Environmental Protection | Environmental Protection Law No. 6 of 2017 | Establishes the Ministry of Environment as responsible for regulations on environmental protection including facility permits, waste management, and pollution control. | [89–91] |
| | Law No. 52 of 2006 and Law No. 42 of 2014 | Focuses on environmental protection and sustainable development, combating pollution and promoting resource management. | [89–91] |
| | Regulation No. 68 of 2020 | Manages hazardous materials and waste according to Environmental Protection Law No. 6 of 2017. Imposes stricter penalties for improper waste disposal. | [89–91] |
| Sustainable Consumption and Production | National Action Plan for Sustainable Consumption and Production (NAP-SCP) | Addresses sustainable practices in agriculture, transportation, and waste management sectors, aligning with SDGs and promoting comprehensive development. | [4–6,98–100] |
| | Law No. 8 of 2022 on Public Procurement | Ensures transparency, fairness, and competition in government procurement, potentially incorporating environmental considerations. | [89,90] |
| | Renewable Energy and Energy Efficiency Law No. 13 of 2012 | Promotes renewable energy sources, exempting related systems from customs duties and sales tax, and easing project implementation processes. | [89,90] |
| Climate Change | Climate Change Law No. 79 of 2019 | Sets an institutional and regulatory framework for climate change activities, including emission regulations and institutional arrangements. | [2,4–6,89,90] |

## 5. Results, Discussion, and Limitations

SDGs are not legally binding, but countries are expected to take ownership and establish a national framework for achieving the 17 goals. Therefore, more action is needed to achieve sustainability in Jordan. Regulation and legislation can be a pillar for sustainability and a powerful force in driving sustainable practices. It can help to reduce environmental risks and promote sustainability by addressing the linkages between society, the environment, the economy, regulation, and sustainability. Therefore, regulation can be an important pillar for sustainability by setting standards and incentivizing sustainable practices. Governments around the globe are increasingly issuing regulations, guidance, and incentives related to sustainability, and environmental and social governance. However, government regulations and legislations alone are not enough to achieve the SDGs. They must be accompanied by effective policies, programs, and partnerships that involve all stakeholders, including governments, civil society, and the private sector. Also, environ-

mental inspection and control are important components of ensuring compliance with environmental laws and regulations.

Jordan has made progress toward achieving some of the SDGs, but more needs to be done to meet the targets by 2030, and actions more than just regulation and legislation are needed. Therefore, while regulation and legislation are important components of achieving sustainability in Jordan, they must be accompanied by other measures such as efficient implementation, comprehensive and evolutionary reforms, commitment to sustainability measures, and sector-specific approaches. GHRM and GSCM, with the enforcement and incentives of regulations and legislation, can help Jordan achieve the SDGs, such as decent work, responsible consumption and production, and climate action.

The Voluntary National Reviews (VNRs) of Jordan provide insights into the country's progress toward achieving the SDGs. While the VNR reports primarily focus on the overall progress of Jordan in achieving various SDGs, we can identify some key links between GHRM/GSCM/regulation and legislation and SDGs 8, 12, and 13 [4,5]. Table 7 provides a concise overview of how SDGs 8, 12, and 13 are interconnected with GHRM/GSCM/regulation and legislation in supporting sustainability efforts.

**Table 7.** The interconnection of GHRM/GSCM/regulation and legislation with SDGs.

| SDG | Description | Connection with GHRM | Connection with GSCM | Connection with Legislation and Regulation | References |
|---|---|---|---|---|---|
| 8 | Promote sustained, inclusive, and sustainable economic growth, full and productive employment, and decent work for all. | The implementation of GHRM practices to integrate environmental sustainability into HR policies, fostering green job creation and ensuring employee engagement in sustainability goals. | Collaboration with suppliers to adopt eco-friendly practices, reducing environmental impact across the supply chain. | The development and implementation of legislation and regulations supporting green job creation, labor rights, and sustainable economic policies. | [4–8,30,44,50,85] |
| 12 | Ensure sustainable consumption and production patterns, including resource efficiency, waste reduction, and sustainable procurement. | The adoption of GHRM initiatives to promote responsible consumption and production among employees, incorporating environmental awareness and sustainable practices into HR policies. | The implementation of green procurement strategies, sourcing materials from sustainable suppliers, and reducing waste generation in the supply chain. | The enactment of legislation and regulations promoting sustainable procurement, waste reduction, and product lifecycle management to achieve responsible consumption and production goals. | [4–6,12,13,55,63,87,100] |
| 13 | Take urgent action to combat climate change and its impacts, including reducing greenhouse gas emissions, increasing climate resilience, and transitioning to low-carbon economies. | The integration of climate-related training and development programs into GHRM practices to educate employees on climate action and reduce carbon footprints. | The implementation of green supply chain practices to reduce emissions, optimize energy use, and promote renewable energy sources throughout the supply chain. | The establishment of legislation and regulations supporting climate action, including emission reduction targets, renewable energy mandates, and carbon pricing mechanisms to incentivize sustainable practices and mitigate climate change impacts. | [4–6,52,53,98–100] |

Implementing GHRM and GSCM practices can help businesses become more environmentally conscious, improve their sustainable performance, and gain a competitive advantage. By adopting these practices, businesses can reduce their environmental impact, enhance their long-term values, and contribute to a more sustainable future. A brief summary of the GHRM and GSCM practices and how they interact with SDGs is shown in Table 8.

**Table 8.** The green practices of HRM and SCM connected to the SDGs.

| Dimensions and Practices | SDGs | | | | | | | | | | | | References |
|---|---|---|---|---|---|---|---|---|---|---|---|---|---|
| | 1 | 3 | 4 | 7 | 8 | 9 | 11 | 12 | 13 | 14 | 15 | 17 | |
| GHRM | | √ | √ | | √ | | | √ | √ | | √ | √ | [3,14–16,30,31,34–37,41–47] |
| Green recruitment and selection | | | | | √ | | | √ | √ | | √ | | [4,31,35,36,41] |

**Table 8.** *Cont.*

| Dimensions and Practices | SDGs | | | | | | | | | | | | References |
|---|---|---|---|---|---|---|---|---|---|---|---|---|---|
| | 1 | 3 | 4 | 7 | 8 | 9 | 11 | 12 | 13 | 14 | 15 | 17 | |
| Green compensation management | | √ | | √ | √ | | | √ | √ | | | | [4,34–36,42,44] |
| Green training and development | | | √ | | √ | | | √ | √ | | | √ | [3,31,34–36,41,42,45,46] |
| Green performance management | | | | √ | √ | | | √ | √ | | √ | | [4,31,35,36,42] |
| GSCM | √ | | | √ | √ | √ | √ | √ | √ | √ | √ | √ | [3,14,20,22,65–83,86] |
| Green purchasing | | | | √ | √ | √ | | √ | √ | √ | √ | | [20,65,67,69–73] |
| Internal environmental management | | | | | √ | √ | | √ | √ | | √ | √ | [74–77] |
| Green logistics | | | | | √ | √ | √ | √ | √ | √ | | √ | [68,78–83] |
| Green manufacturing and packaging | √ | | | √ | √ | √ | | √ | √ | | | | [8,69,82,84,85] |

From the table above, it can be seen that all practices of GHRM and GSCM achieve SDG8, SDG12, and SDG13. However, it does not only achieve those, but it also achieves other SDGs, such green manufacturing and packaging related to SDG1 and green compensation management related to SDG3. Furthermore, the research shows how the dimensions and practices will achieve the three pillars of sustainability (see Table 9). Future research can investigate the relation between the practices and other goals.

**Table 9.** The relation between sustainability pillars and practices.

| Dimensions and Practices | Social Pillar | Economic Pillar | Environmental Pillar |
|---|---|---|---|
| Green recruitment and selection | | √ | √ |
| Green compensation management | √ | √ | √ |
| Green training and development | √ | √ | √ |
| Green performance management | √ | √ | √ |
| Green purchasing | √ | √ | √ |
| Internal environmental management | | √ | √ |
| Green logistics | √ | √ | √ |
| Green manufacturing and packaging | √ | √ | √ |

The investigation into the development of GHRM, GSCM, and the existing regulatory and legislative framework in Jordan with the overarching aim of achieving the SDGs has yielded valuable insights and raised critical considerations.

The findings suggest that the integration of GHRM and GSCM practices can significantly contribute to the advancement of the SDGs in Jordan. Organizations that adopt sustainable human resource and supply chain practices tend to exhibit positive impacts on environmental preservation, social equity, and economic sustainability. The analysis further indicates that a proactive approach to aligning these practices with the existing regulatory and legislative landscape is crucial for fostering a sustainable business environment.

Based on the findings of this study, it is recommended that organizations in Jordan adopt and implement GHRM and GSCM practices to improve their environmental performance. This can include practices such as green training and development, green performance management, green purchasing, and green logistics. Furthermore, it is important for the government to continue to enforce regulations and legislation that promote sustainable development and protect the environment. This can include measures such as promoting investment in renewable energy and ensuring equal pay for work of equal

value. Organizations can also prioritize green recruitment and onboarding practices to facilitate the adoption of GSCM practices. Additionally, incorporating green training into organizational policies can improve employee commitment toward the environment and contribute to sustainable workplace practice.

The discussion reveals several key pathways for the development of GHRM, GSCM, and regulatory and legislative frameworks in Jordan to better support the achievement of the SDGs. Collaboration between businesses, government bodies, and non-governmental organizations is essential for creating a conducive environment for the implementation of sustainable practices. Additionally, the promotion of awareness and education regarding the benefits of GHRM and GSCM can play a pivotal role in fostering a culture of sustainability among stakeholders. Strategic partnerships between the private sector and regulatory bodies are identified as a potential avenue for driving policy reforms that align with international sustainability standards. The discussion underscores the importance of creating incentives and recognition mechanisms to motivate organizations to adopt and adhere to green practices, enhancing their commitment to sustainable development.

Despite the strengths of this study, there are additional limitations that should be acknowledged. Firstly, the research is inherently context-specific to Jordan and may not be directly applicable to other regions with distinct socio-economic, cultural, and regulatory landscapes. Additionally, the dynamic nature of regulatory environments and business practices introduces an element of temporal sensitivity, emphasizing the need for continuous monitoring and adaptation. This study's scope may not comprehensively cover all potential variables influencing the development of GHRM, GSCM, and regulatory frameworks in Jordan. As such, further research is warranted to explore additional factors that may impact the successful implementation of sustainable practices. Furthermore, it would be valuable to explore other dimensions that may influence green entrepreneurial activity and sustainable development. Secondly, this study relied on secondary data for the period 2016–2023 and from a specific database; future studies should consider a wider time span and more databases to enable long-term analyses and enhance overall knowledge. Such an approach will have dynamic effects, revealing different or similar entrepreneurial responses when institutional factors change in developing countries.

Based on the identified limitations, future research should aim to delve deeper into the intricacies of stakeholder collaboration and assess the effectiveness of specific incentive structures in promoting sustainable practices. Comparative studies across different industries and regions could provide a more nuanced understanding of the diverse challenges and opportunities associated with the development of GHRM, GSCM, and regulatory frameworks. It would be worthwhile for future studies to extend the analysis to include cross-country comparisons, such as examining other regions within the Middle East and North Africa (MENA) region. Future research should use another approach and another environment to improve the quality and scope of the indicators, for both dependent as well as independent variables. The limitations of this study include the need for further empirical evidence and critical analysis to fully understand the interconnectedness between green supply chain management, green human resource practices, and government regulations in promoting sustainable development in Jordan. Additionally, this study acknowledges the need for more research to address the identified limitations and to provide a more comprehensive understanding of the impact of these factors on SDGs 8, 12, and 13. Furthermore, this study could benefit from more in-depth case studies and real-world examples to demonstrate the practical application and effectiveness of the proposed conceptual framework. These limitations highlight the potential for future research to build upon the current findings and provide a more robust understanding of the subject matter.

In conclusion, this study highlights the pivotal role of GHRM, GSCM, and regulatory and legislative frameworks in driving sustainable development in Jordan. By addressing the identified limitations and building upon the insights gained, future endeavors can further refine strategies for fostering a holistic and effective approach to achieving SDGs through integrated green practices.

**Author Contributions:** L.F.: first draft production, research, data collection, and editing; M.A.-Q.: research, data collection, and editing; Z.H.: supervision and reviewing the text; M.A.: supervision and reviewing the text. All authors have read and agreed to the published version of the manuscript.

**Funding:** This research received no external funding.

**Institutional Review Board Statement:** Not applicable.

**Informed Consent Statement:** Not applicable.

**Data Availability Statement:** No new data were created or analyzed in this study. Data sharing is not applicable to this article.

**Conflicts of Interest:** The authors declare no conflicts of interest.

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
