# Peer review of "Green Human Resource Management/Supply Chain Management/Regulation and Legislation and Their Effects on Sustainable Development Goals in Jordan"

_sustainability, doi:10.3390/su16072769_

Round 1
Reviewer 1 Report
Comments and Suggestions for Authors
The objective of this article is to propose a conceptual framework of the research model included the correlations between Government regulations and legislations, GHRM, GSCM and Sustainable Development Goals (SDG). The subject matter is current and consistent with contemporary trends in the development of the world economy.
The authors conducted a satisfactory review of the literature (the list of literature contains 100 items, of which over 70 items were published after 2019). I assess Table 1, which is one of the results of this review, positively. This is a very clear presentation of data, the review may not be complete, but it is fully satisfactory in terms of form and substantive value.
The research methodology is presented in a clear way and contains all the information needed to understand the authors' research intention and the actions they have undertaken. A conceptual framework of the research is a valuable supplement to this part of the article. In general, I evaluate the graphic elements and tables positively, they are clear and satisfactory in terms of content. The only exception to this rule is Table 3, which seems unnecessary. You can replace it with plain text.
There are some minor stylistic errors, such as "the three pillars of green HRM can be 177 summarized for three". Additionally, some chapter titles use unnecessary ":" characters and the title of chapter 4.4.2 Economic Diversification is located in the text. Re-reading the text will definitely help remove these errors.
The biggest weakness of the article is the insufficient emphasis on the authors' own contribution. It is important that authors:
1. Determined what are the benefits of using their model and how does it differ from others available in the literature?
2. Identified the main achievements from the use of their model. The authors should indicate the implications of the relationships they have identified (currently, the data included in the summary are devoid of more valuable commentary from the authors). What is the element of novelty compared to the research results of other authors?
3. What are the limitations of the research methodology used? What are the possibilities for its further development?
In general, the topics contained in the article are interesting and up-to-date. I am convinced that after the authors make corrections, the article will have scientific value and will be an interesting read for people interested in this topic.
Dear authors, good luck!
Author Response
Thank you very much for taking the time to review this manuscript. Please find the detailed responses below and the corresponding revisions.
All the new changes made to the manuscript are written in yellow highlighted to make it easier for the reviewers to track them.
- Comment 1: The researchers added the benefits and how does it differ from others available in the literature to the abstract, as stated in the manuscript's comments and highlighted in yellow.
- Comment 2: The researchers added it to the Abstract and Results, Discussion, and Limitations, as stated in the manuscript's comments and highlighted in yellow.
- Comment 3: The researchers added it to the Abstract and Results, Discussion, and Limitations, as stated in the manuscript's comments and highlighted in yellow.

Reviewer 2 Report
Comments and Suggestions for Authors
Green Human Resource Management/Supply Chain Management /Regulation and Legislation and its effects on Sustainable Development Goals in Jordan
Here below is my review report summary on the title mentioned above.
1. Despite being well-prepared and of good quality, the abstract could benefit from including additional information concerning the study methodology, the unique aspects of the research, and any recommendations made by the authors.
2. The introduction section is scanty, the authors should include some empirical studies conducted on Green Human Resource Management/Supply Chain Management /Regulation and Legislation and its effects on Sustainable Development Goals. At the same time, more emphasis should be given to Green Human Resource Management/Supply Chain Management /Regulation and Legislation than SDGs.
3. The literature section is not convincing. First of all, the authors should discuss some established theories, and empirical evidence on the proposed topic, then the authors can summarize their results and overall assumption in the table form.
4. The authors should mention the objective of this study, the motivation behind conducting this study, and the novelty of the study.
5. The methodology section is not clear and needs to be modified. What were the databases used? What criteria were used for inclusion and exclusion? Is this study a literature review? If yes, which type of review? Systematic, bibliometric, meta-analysis, or rapid review? Where is the PRISMA flow diagram chart? The study title and the methodology contradict each other.
6. The conceptual framework is not convincing and is not consistent with the title.
This manuscript underwent a thorough evaluation, receiving many specific comments that proved crucial in shaping its pathway to publication. Given the insightful feedback received, I would like to propose a constructive suggestion to the authors. It would greatly enhance the manuscript's depth and credibility if the authors delve into existing studies that have employed a similar methodology. Additionally, the literature section, as it stands, appears somewhat limited; a more comprehensive review of studies is imperative to derive up-to-date conclusions regarding the study. Moreover, it is noteworthy that the methodology employed in this study differs significantly from its title. To bolster the transparency and coherence of the research, the authors should meticulously elucidate the methodology adopted, highlighting the specific use of literature review, which was deemed advantageous for this study. Further clarification on databases is also needed. As it stands, the manuscript can be rejected right away, but let us give second chance if it can improve.

Minor editing
Author Response
Thank you very much for taking the time to review this manuscript. Please find the detailed responses below and the corresponding revisions.
All the new changes made to the manuscript are written in yellow highlighted to make it easier for the reviewers to track them.
- Comment 1: The researchers added the methodology, the unique aspects of the research, and any recommendations made by the authors to the abstract, as stated in the manuscript's comments and highlighted in yellow.
- The introduction section is scanty, the authors should include some empirical studies conducted on Green Human Resource Management/Supply Chain Management /Regulation and Legislation and its effects on Sustainable Development Goals. At the same time, more emphasis should be given to Green Human Resource Management/Supply Chain Management /Regulation and Legislation than SDGs.
The researchers emphasis the main concepts of the study and narrow down only the main idea, the researcher added the aims and main objectives to the introduction. More emphasis were given to Green Human Resource Management/Supply Chain Management /Regulation and Legislation appeal in the literature review and the emphasis of SDGs were only in introduction and small parts from literature.
- Comments 3: The researchers merge two section. The first part described literature review of the variable and the second described literature review of practices. The researchers added discussion of some established theories, and empirical evidence on the proposed topic, as stated in the manuscript's comments and highlighted in yellow.
- The authors should mention the objective of this study, the motivation behind conducting this study, and the novelty of the study.
The researcher added the objectives, motivations and novelty to abstract and introduction, as stated in the manuscript's comments and highlighted in yellow.
- The methodology section is not clear and needs to be modified. What were the databases used? What criteria were used for inclusion and exclusion? Is this study a literature review? If yes, which type of review? Systematic, bibliometric, meta-analysis, or rapid review? Where is the PRISMA flow diagram chart? The study title and the methodology contradict each other.
The researchers added the data bases and the criteria were used for inclusion and exclusion also we designed a flow diagram for the design and methodology, as stated in the manuscript's comments and highlighted in yellow.
- The conceptual framework is not convincing and is not consistent with the title
The researchers redesign the conceptual framework to be consistent with the title

Round 2
Reviewer 1 Report
Comments and Suggestions for Authors
All the elements I had reservations about have been corrected. I have no further comments.
Author Response
Thank you
Reviewer 2 Report
Comments and Suggestions for Authors
All my concerns are not responded properly. I still reccomend to revist my previous comments
Author Response
Dear Reviewer,
I appreciate your assistance in reviewing the manuscript titled “Green Human Resource Management/Supply Chain Management/Regulation and Legislation and its effects on Sustainable Development Goals in Jordan”, by Lana Freihat, Mousa Al-Qaaida, Zayed Huneiti, Maysam Abbod.
In the following section, we outline how the comments have been taken into account and incorporated into the revised version of the article, the yellow highlighted text that indicates the changes made in the previously revised manuscript.
In addition, all the new changes made to the manuscript are written in yellow highlighted to make it easier for the reviewers to track them.
Reviewer comments
The researchers enhanced the abstract by incorporating additional information on the study methodology, unique research aspects, and any recommendations made by the authors.
They also included some empirical studies in the introduction section, along with added aims and questions that describe the relationship. Emphasis was given to Green Human Resource Management, Supply Chain Management, and the impact of Regulation and Legislation on Sustainable Development Goals.
The literature section was strengthened by discussing established theories and empirical evidence related to the proposed topic. The results and assumptions were summarized in a table.
The study's objectives, motivation, and novelty were included in the manuscript. Additionally, aims and questions describing the relationship were incorporated.
The methodology section was clarified and modified, specifying databases used, inclusion/exclusion criteria, and the review type. The methodology model was updated to include a review of regulations and law judicial decisions.
Results, discussion, and limitations were also added to the last section.
The conceptual framework was revised to make it more convincing and consistent with the study title.
With my best regards,
Dr. Mousa Al-Qaaida
Assistant Professor of Public Law
Amman Arab University / Jordan
Email: m.alqaaida@aau.edu.jo / mosa.sami87@hotmail.com

Round 3
Reviewer 2 Report
Comments and Suggestions for Authors
Thank you for your valuable feedback on the manuscript. While the authors have generally done a praiseworthy job, I do have reservations, particularly regarding the methodology section of the study.
The authors claim to have employed a systematic review for this study, but I find that the reporting falls short of the standard systematic review process. Notably, there is a lack of mention of inclusion and exclusion criteria, and the absence of a PRISMA flow chart diagram (http://prisma-statement.org/prismastatement/flowdiagram.aspx?AspxAutoDetectCookieSupport=1) raises concerns about transparency in the study's methodology. Additionally, the specific databases utilized for the review (such as Web of Science, Scopus, ScienceDirect, or Google Scholar) are not explicitly stated. It is essential to provide clarity on these aspects to enhance the reproducibility and reliability of the study. (Please refer to this manuscript, https://doi.org/10.3390/su14127416 )
Furthermore, the rationale behind choosing the years 2016 and 2023 is not justified in the manuscript. Providing insight into the reasoning behind this time frame would contribute to a more robust methodology. The keywords used for the search engine are mentioned, but their relevance to the study's title could be more convincingly established.
A critical aspect that needs improvement is the manuscript's reporting layout. I recommend aligning the submission layout with the specific guidelines outlined by the journal. Adhering to the prescribed format will enhance the overall presentation and readability of the manuscript.
While I appreciate the significance of conducting a literature review to identify future research areas, the methodology must be scientifically sound. A thorough and transparent reporting of the systematic review process is essential to prevent potential biases and to encourage trust among academic readers. I believe addressing these concerns will contribute to the overall quality and credibility of the study.
Comments on the Quality of English Language
Minor Editing
Author Response
Thank you for your valuable feedback. We appreciate your thorough review and have taken your suggestions seriously. In response to your concerns, we have updated the manuscript to include a PRISMA flow chart diagram, explicitly stated the databases used (Scopus and Google Scholar), and provided justification for choosing the years 2016 and 2023. Additionally, we have aligned the submission layout with the specific guidelines outlined by the journal for enhanced presentation and readability. We hope these revisions address your concerns and contribute to the reproducibility and reliability of the study.
We have duly highlighted the revisions and updates in the manuscript to ensure transparency and draw attention to the improvements made in response to your constructive feedback.

Round 4
Reviewer 2 Report
Comments and Suggestions for Authors
Congratulations. Well done!
Comments on the Quality of English LanguageMinor editing
Author Response
Revision of Manuscript ID Sustainability-2810516
Dear Reviewer
I am writing to submit the revised manuscript entitled Green Human Resource Management/Supply Chain Management/Regulation and Legislation and its effects on Sustainable Development Goals in Jordan. I would like to express my gratitude for the valuable feedback provided by the reviewers, which greatly contributed to improving the quality and comprehensiveness of our manuscript.
We have integrated new content into Green Human Resource Management (GHRM), Green Supply Chain Management (GSCM), and Government Regulations and Legislations sections to provide a more comprehensive overview of the topic. Specifically, we have included Table 4 summarizing examples of GHRM practices, Table 5 summarizing examples of GSCM practices, and Table 6 outlining the laws and regulations relevant to various aspects of sustainable development in Jordan. These changes have been highlighted with yellow for the editors' and referees' attention.
Additionally, within the Results, Discussion, and Limitations sections, we have included Table 7 providing a concise overview of how SDGs 8, 12, and 13 are interconnected GHRM/GSCM/Regulation and Legislation supporting sustainability efforts. The changes made to this table and the accompanying discussion have also been highlighted with yellow for the editors' and referees' attention.
Additionally, we have made changes to some sentences throughout the manuscript to improve clarity and readability, ensuring that the revisions are highlighted with yellow for the editors' and referees' attention.
